# Development of Performance Evaluation Indicators for Table Grape Packaging Units. 2. Global Indexes

**Edson Kogachi \*** , **Adonias Ferreira, Carlos Cavalcante and Marcelo Embiruçu**

Industrial Engineering Program, Polytechnic Institute, Universidade Federal da Bahia, Salvador 40210630, Brazil; adoniasmagdiel@ufba.br (A.F.); arthurtc@ufba.br (C.C.); embirucu@ufba.br (M.E.)

**\*** Correspondence: ekogachi@gmail.com

**Abstract:** The adoption of a global index (GI) for performance evaluation has been increasingly recognized as a useful strategy for decision-making as it simplifies the interpretation and monitoring of the results. Because the GI is often built by adopting a combination of different procedures for normalization, weighting, and aggregation of indicators, it is challenging to select the optimal combination of procedures, since the countless combinations lead to different results. This paper proposes a method for the development of a robust and original GI for the evaluation of table grape production units (TGPUs). Various combinations of procedures were used to develop eighteen GIs for each TGPU. These are located in the lower-middle San Francisco valley in the northeast of Brazil, where their robustness was assessed by identifying outlier GIs and via a graphical analysis. Plausible GIs were reliably identified and a cluster analysis was conducted to categorize the TGPUs into groups considering each performance objective. The identification of the outlier GIs and the use of the plausible GIs in cluster formation constitute a new scientific approach to the topic, which can be extended to other applications and contribute to the sustainable development of several industries.

**Keywords:** global index; cluster analysis; packaging; table grapes; sustainability; lower-middle San Francisco valley in Brazil

## 1. Introduction

Several previous studies have addressed the definition, development, and use of indicators for various objectives, such as performance evaluations, action planning, control of achievements, and positioning in relation to global and strategic objectives, in many areas [1]. An increasing challenge is the proposal of indicators that encompass several dimensions in a single indicator, known as a global index (GI). GIs have been increasingly recognized as a useful tool [2] and several studies have emphasized the need to assess performance using a GI [3–6]. Examples of GIs used in multidimensional performance assessment include the human development index (HDI) and gross domestic product (GDP). They are considered easier to interpret than a set of individual indicators, easier to use for educating and communicating to the general public, easier to monitor progress over time, and more useful for decision-making [7].

Among the papers addressing the development of GIs, those that adopt normalization, weighting, and aggregation procedures make up the majority [8–10]. The observed variations in the results of these methods are due to the many possible combinations of these procedures (normalization, weighting, and aggregation). Therefore, it is necessary to analyze the combinations of procedures because the heterogeneity of the results reduces the credibility of the method [1]. If the GI is poorly constructed, it can be used to send misleading messages or even to manipulate results in favor of certain interests [11].

Some authors [2] have expressed concerns that the heterogeneity of the combinations reduces the credibility of the GI used to rank the performance of countries in areas such as industrial competitiveness, sustainable development, globalization and innovation.

They have proposed that the robustness of the GI should be assessed considering how the selection of the input procedures propagates through the structure of the GI and affects the results. However, although robustness analysis promotes greater transparency in the design and increases the credibility of the GI, developers have given this little importance and often skip this verification all together [12].

The GI is an interdisciplinary tool applied in almost all research areas [13]. This paper develops and applies GIs to the production process of table grape packaging units (TGPUs) in an agricultural region that focuses on the exportation of grapes. TGPUs are considered a critical item in the table grape supply chain [14–16]. They are also known as packing houses, which are suitable facilities that receive the grapes harvested in the field and perform the cleaning, classification, and packaging processes [17]. Such TGPUs exist in several countries and have a seasonal and labor-intensive work environment due to the special care required to avoid damage when handling the fragile fruit, which is only possible with manual labor [18].

Previous works have discussed GIs in the performance evaluation of agricultural processes [8,10,19], however, the robustness of the GIs has not been analyzed in this context. This gap in the scientific literature motivated this study: the development of a robust GI for the performance evaluation of TGPUs. The method proposed in this research was verified for TGPUs located in the *vale do submédio São Francisco* (VSSF; San Francisco lower middle valley), in northeast Brazil, using previously developed performance indicators [18]. The structure of the proposed method for developing a robust GI for TGPUs is illustrated in Figure 1.

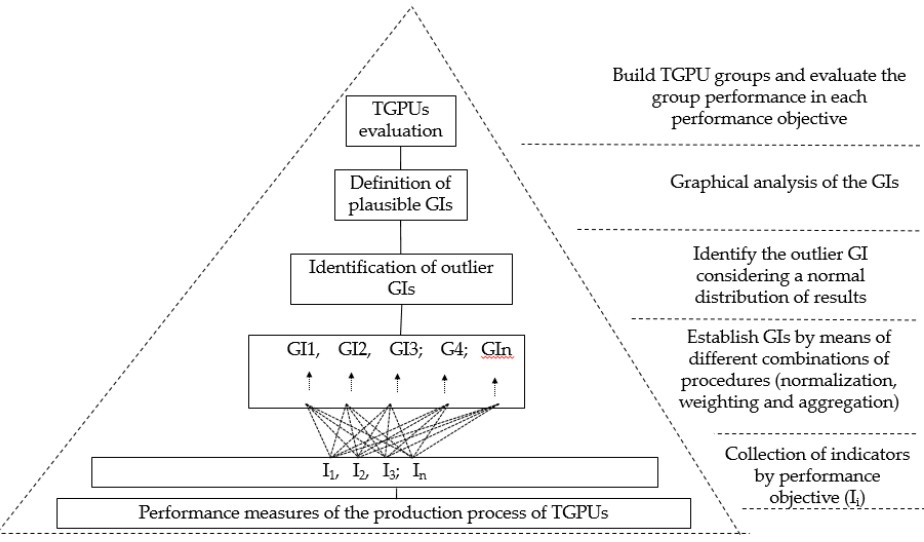

**Figure 1.** Research structure for developing GIs for TGPUs.

The background of GIs and the methods used here are presented in Sections 2 and 3, respectively. A description of the TGPUs, the application of GIs for sustainability topics and the study hypotheses are shown in Section 4. The case study of VSSF grape production and the calculations of the GIs are discussed in Section 5. Finally, conclusions and suggestions for future work are provided in Section 6.

## 2. Global Indexes (GIs) Background

The term GI has several other labels, such as composite indicator (CI), integrated indicator, or multidimensional indicator. The increasing reduction from raw data to indicators, and finally to indexes, represents the hypothetical progression of measurements [20]. Indexes can be constructed from analyzed data by aggregating a set of data elements with established relationships [21]. The index (or indexes) is simply a high-order indicator and is an aggregate or weighted cluster of indicators [22]. In this work the term GI is used to

refer to the result of the aggregation of indicators. Figure 2, adapted from [23], illustrates a pyramid relating the raw data, indicators, and the GI as the level of aggregation increases.

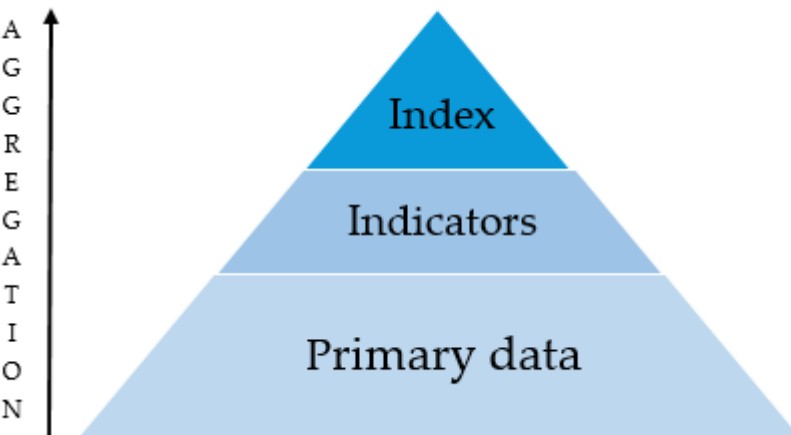

**Figure 2.** Relationship between primary data, indicators, and the final index.

Sometimes GIs are developed using specific methods, such as data envelopment analysis (DEA), principal component analysis (PCA), and factor analysis (FA). However, most often they are formed by the normalization, weighting, and aggregation of indicators. Table 1 describes the scientific publications from the last 10 years (2011–2020) related to the application of normalization, weighting, and aggregation procedures in the GI composition for different scopes of evaluation. The compiled references (in Table 1) highlight the min-max normalization, participative and equal weightings, and linear aggregation. In addition, the references mostly cover scopes related to sustainability in lato sensu (environmental, social and sustainability in stricto sensu), in which data are frequently provided by government agencies.

The utility, quality, and reliability of a GI strongly depend on the combination of normalization, weighting, and aggregation procedures, where different combinations give different results [24]. Considering the wide range of methodologies applied in the development of GIs and the growing interest in their use, a manual was developed by the Joint Research Center of the European Union and the Organization for Economic Cooperation and Development [2] to compare and rank the performance of countries in areas such as industrial competitiveness, sustainable development, globalization, and innovation. The objective was to provide researchers with a set of recommendations on how to design, develop, and disseminate a GI. The authors suggested a checklist with ten items to be followed: (1) theoretical framework; (2) data selection; (3) imputation of missing data; (4) multivariate analysis; (5) normalization; (6) weighting and aggregation; (7) robustness and sensitivity; (8) return to the data; (9) links to other variables; and (10) presentation and visualization.

**Table 1.** Procedures (normalization, weighting and aggregation) to compose GI in different scopes of evaluation.

| Reference | Scope of Evaluation | Normalization | Weighting | Aggregation |
|---|---|---|---|---|
| [8] | AGR | Min–Max (0–1) | Participative (AHP) | Linear |
| [9] | AGR | Min–Max (0–1) | Participative (AHP) | Linear |
| [25] | AGR | Min–Max (0–1) | Participative | Linear |
| [5] | AGR | Min–Max (0–1) | Participative (AHP) | Linear |
| [26] | ENV | Min–Max (0–1) | Equal | Linear |
| [27] | ENV | Min–Max (0–100) | Participative | Linear |
| [28] | ENV | Min–Max (0–1) | Participative (AHP); Statistic (PCA) | Linear; Geometric |
| [29] | ENV | Relative to the maximum; Z-scores | Participative; Statistic (Factor analysis) | Linear |
| [30] | ENV | Relative to the maximum | Equal | Linear; Geometric |
| [31] | ENV | Min–Max (0–1) | Statistic (Entropy) | Linear |
| [32] | ENV | Min–Max (0–1) | Statistic (Entropy) | Linear |
| [33] | ENV | Min–Max (0–100) | Equal | Linear |

**Table 1.** *Cont.*

| Reference | Scope of Evaluation | Normalization | Weighting | Aggregation |
|---|---|---|---|---|
| [34] | INO | Relative to the maximum | Equal; Statistic (DEA, PCA, FA) | Linear |
| [35] | SOC | Min–Max (0–1) | Equal | Linear |
| [36] | SOC | Min–Max (1–100) | Participative (AHP) | Geometric |
| [37] | SOC | Relative to the median | Equal | Linear |
| [38] | SUS | Min–Max (0–1) | Equal | Linear |
| [39] | SUS | Min–Max (0–1) | Participative (AHP) | Linear |
| [40] | SUS | Z-scores; Min–Max (0–1); Borda count; Relative to the maximum and mean | Equal | Linear; Geometric; Concave mean |
| [11] | SUS | Z-scores; Min–Max (0–1); Borda count; Relative to the maximum and mean | Equal | Linear; Geometric; Concave mean |
| [41] | SUS | Min–Max (0–1) | Participative (AHP) | Linear |

SUS = sustainability; ENV = environmental; SOC = social; AGR = agriculture; INO = innovation; AHP = analytic hierarchy process.

This checklist was used as a guideline for this study, however, the first four of the ten suggestions will not be covered in this paper because they have already been addressed in our proposal for the development of performance indicators for production processes of TGPUs [18]. In addition, item 9 ("links to other variables") was not considered due to the absence of another GI that can be correlated to the one developed here. Hence, the remaining five suggestions addressed in this study were classified into three topics: (i) development of GIs in this study (items 5 and 6, normalization, weighting, and aggregation); (ii) checking robustness of the GIs (item 7); (iii) techniques related to the clustering of TGPUs (items 8 and 10, back to the actual data and presentation and visualization).

## 3. Methods for the Development of GIs

### 3.1. Normalization

Generally, the indicators are measured in different measurement units and in various ranges and value scales and therefore they need to be positioned on a common baseline to avoid problems introduced by the different measurement units [42]. Following [43], the indicators were standardized to avoid the dominance of extreme values over the others, and to partially resolve low-quality data problems. The concepts inherent in some normalization procedures, which are often used in GI development, are described below in accordance with [26].

1.  Classification (Borda count): normalizes the indicators using the values rank or classification. This simple normalization technique is unaffected by discrepant data points and allows the performance to be tracked over time in terms of relative positions (rankings). However, the absolute level of the element under evaluation is compromised because information about the differences in performance between the evaluated elements is lost.

2.  Z-scores: converts indicators into a common scale with zero mean and one standard deviation. In this normalization, indicators with extreme values have a greater effect on the GI. The extreme values influence the results because the range between the minimum and maximum standardized scores varies for each indicator, thus enhancing the value of the item under evaluation that has more extreme values.

3.  Min-Max: standardizes the indicators to achieve an identical range, for example between [0, 1], by subtracting the minimum value and dividing by the range of the extreme values. This normalization is based on scaling and not on standard deviation. While the method may be more robust when there are many discrepant values, if there is little variation in the values the normalization will extend the indicator ranges to extreme values.

4.  Relative to the maximum value: assigns a value of 1 to the highest value of a specific indicator, while the other values are classified as a fraction of the maximum. Therefore,

the closer the value is to the maximum, the closer it is to 1. This can enhance discrepant values that are far from the maximum.

5.  Relative to the mean/median value: assigns a value of 1 to the chosen reference value of a specific indicator, for example the median, and therefore values above the reference receive a value higher than 1 and the smallest ones receive values below 1. Statistically, this process is more vulnerable to the influence of discrepant values than other procedures.

The different normalization procedures do not affect the classification of the evaluated indicator values because they are simple ranks or linear transformations. However, they do affect the GI and consequently the classification of the item under evaluation since the individual standardized indicators are also aggregated to form the GI. These normalization procedures were chosen and adopted in this work as they are commonly used for developing GIs and their respective equations are shown in Table 2.

**Table 2.** Normalization methods.

| Method | Equation |
|---|---|
| Classification (Borda count) | $I_e^i = 1 - \frac{R(x_e^i) - 1}{n}$ |
| Z-scores | $I_e^i = \frac{x_e^i - \widetilde{x}^i}{\sigma^i}$ |
| Min–Max | $I_e^i = \frac{x_e^i - (x^i)}{(x^i) - (x^i)}$ |
| Relative to the maximum value | $I_e^i = \frac{(x^i) - |(x_e^i - (x^i))|}{(x^i)}$ |
| Relative to the mean/median value | $I_e^i = \frac{\widetilde{x}^i + (x_e^i - \widetilde{x}^i)}{\widetilde{x}^i}$ |

$I_e^i$ = the standardized indicator for the variable $i$ and unit $e$; $R$ = the rank; $n$ = the total number of production units; $x_e^i$ = the indicator for $i$ and $e$; $\widetilde{x}^i$ = the mean/median of $i$; $\sigma^i$ = the standard deviation of $i$; $x^i$ = the indicator for $i$.

### 3.2. Weighting and Aggregation

The weights for the different performance indicators are essentially value judgments about relative importance. The weights can drastically alter the units ranking if an indicator is weighted highly and a given unit has a high positive or negative value in this specific indicator. The weighting can rely on statistical methods [28,29,31,32,34], including those based on the standard deviation of the indicators or the correlations between them [44,45]. Participative methods are also used, where expert opinions are used to reward or punish components considered more or less influential [5,8,9,25,27–29,36,39,41]. Although weighting can be subjective, there are valid reasons for using it.

Frequently, equal weights are adopted in the weighting process, which implies that all factors are equally important in the GI [2]. In fact, Table 1 shows that equal weights were used in many performance evaluation studies. According to [42], equal weighting is mainly valid in contexts where statistical or empirical bases are not sufficient to justify the selection of unique weights for each factor. The use of equal weights for the different themes and equal weights among the indicators below each theme has been recommended to avoid imbalances, due to the number of indicators being different in each theme [11]. Considering this background, in our study we use the equal weights, and the weights considering the participation of experts and specialists through AHP in the development of the GIs, as they are the two most used in performance evaluation studies, as shown in Table 1.

The participative method most used to establish criteria weights for indicators is the AHP [5,8,9,28,36,39,41]. AHP breaks down a problem into a hierarchical structure and obtains its weights through paired comparisons between criteria [46]. Participants express their preferences and, through some kind of agreement, consensus, convergence or average, compare which criterion is the most important and classify it on a nine-point scale, ranging from 1 (equally important) to 9 (extremely more important). The construction follows

the steps of building the pairwise comparison matrix and calculations of eigenvector ($V_i$), normalized eigenvector ($V_{in}$) (which is the weights itself), consistency index ($C_i$), and degree of consistency of the matrix ($CR$), in accordance with the equations described in Table 3.

**Table 3.** AHP steps and equations.

| Steps | Equations |
|---|---|
| Pairwise comparison matrix | $A = \left[ a_{i,j} \right]_{n,n}, \quad a_{j,i} = \frac{1}{a_{i,j}}$ |
| Eigenvector | $V_i = \prod\limits_{j=1}^{n} a_{i,j}^{1/n}$ |
| Normalized Eigenvector (weights) | $V_{in} = \frac{V_i}{\sum_{i=1}^{n} V_i}$ |
| Consistency index | $C_i = \frac{\lambda - n}{n-1}, \quad \lambda = \sum\limits_{j=1}^{n} \left( \sum\limits_{i=1}^{n} A_{i,j} \right)_j \cdot V_{jn}$ |
| Degree of consistency | $CR = \frac{C_i}{RI}$ |

$a_{i,j}$ = the importance of indicator (of line) $i$ in relation to indicator (of line) $j$ (1 to 9 or 1/9 to 1); $n$ = the dimension of the matrix (number of indicators or criteria); $RI$ = tabulated values proposed by [46]; $CR \leq 0.1$ is considered acceptable.

The aggregation of indicators takes place after the normalization and weighting steps. As in the previous steps, the use of different aggregation procedures influences the structure of the GI and consequently the classification of the evaluated TGPUs. Although the most recent manual on composite construction divides aggregation methods into linear, geometric, and multicriteria methods, they are all included in the multicriteria decision analysis framework [12]. It has been proposed that the aggregation of indicators to form a composite implies a choice between compensatory and non-compensatory approaches [47]. Each approach is adequate for a specific purpose and involves some advantages and disadvantages.

The compensatory approach involves trade-offs, that is, it allows compensation between criteria [35]. So, if an evaluated unit has an indicator with a low value in one criterion, it can be compensated by an indicator with a high value in another criterion. On the other hand, in the non-compensatory method there are no trade-offs and, in this case, what matters are the comparisons between the pairs of different evaluated units, located in the same criterion. Thus, a greater number of favorable comparisons leads to a better positioning of the evaluated unit [2]. The composition of GIs usually takes the compensatory premise. The GIs for production processes, a theme related to this research, also follow this same premise, namely allowing for compensation among indicators. Hence, this approach is used here.

The compensatory aggregation method most often used in the composition of GIs is simple (linear) aggregation, where compensations are constant. This means that poor performance in one indicator can be fully compensated by good performance in another indicator. Thus, production units with low scores in some variables will benefit from linear aggregation [2]. However, to limit the compensation, geometric aggregation can be used. In this case, if a production unit has a low score in one indicator it will need a much higher score in another indicator to improve its ranking.

Concave-mean aggregation has been proposed as a compromise between linear and geometric aggregation [11], which searches for the aggregate weighted arithmetic mean of a transformation of the standardized indicators [48]. This method rewards performance in a non-linear way, where the reward increases as the relative performance of an evaluated element increases. This means that the imbalances between the different dimensions will have less importance if the relative performances are at moderate to high levels. According to [40], the mean concave aggregation was designed to be applied in normalized indicators between 0 and 1, and in our study it was applied in the normalizations Borda count, min-max, and relative to the maximum.

Among the several procedures available for compensatory aggregation of indicators, this work applied the frequently used ones, as described in Table 4.

**Table 4.** Aggregation methods.

| Method | Equation |
| :---: | :---: |
| Linear | $GI_e = \sum_{i=1}^{m} w_i \, I_{ie}$ |
| Geometric | $GI_e = \prod_{i=1}^{m} (I_{ie})^{w_i}$ |
| Concave mean | $GI_e = \sum_{i=1}^{m} w_i \left( I_{ie} - e^{-I_{ie}} \right)$ |

$GI_e$ = the global index for the production unit $e$; $m$ = the total of indicators; $w_i$ = the weight of the standardized indicator for the variable $i$; $I_e^i$ = the standardized indicator for $i$ and $e$.

### 3.3. Robustness Evaluation

In the case of a system, robustness refers to a design that can accommodate variability of the parameters affecting its performance with acceptable margins of degradation, while achieving the optimal combination of operational costs, reliability, maintainability, and performance [49]. Robustness evaluation is performed in statistics [50], computer science [51], and decision-making [52], the context associated with this work. Robust decision-making processes address the structured planning of complex systems under uncertainties in the input parameters of the model [52]. It seeks to identify robust decisions that satisfactorily assume a wide range of plausible alternatives, rather than an optimal one. Furthermore, robust decision-making further identifies uncertain combinations that contribute to the vulnerability of the systems.

According to [2] it is essential to verify the robustness of a GI during its development. The authors suggest evaluating the chosen procedures because the quality of the GI can lead to a questionable interpretation. They specify that robustness can be assessed through a combination of uncertainty and sensitivity analysis. While the uncertainty analysis focuses on the identification of input factors (selection of individual indicators, data quality, methods of normalization, weighting and aggregation, among others) capable of altering the GI result and, based on that, allows to quantify the overall uncertainty of the GI or CI, the sensitivity analysis assesses the specific contribution of each individual input factor, showing, for example, how much uncertainty in the GI can be reduced (or increased) if a specific source of uncertainty is decreased (or increased). Therefore, the sensitivity analysis is the study of how the uncertainty in the output can be apportioned to the different sources of uncertainties in the input variables or, to put it another way, how the variability in each input factor affects the results and the uncertainty associated with the GI, thus being a complementary study of the uncertainty analysis. Because of this, these combined procedures help in assessing the robustness of the GI.

Studies have been carried out on the verification of GI robustness using various approaches, which consider the uncertainties related to the use of different normalization, weighting, and aggregation procedures. The GIs developed by the different procedures were graphically evaluated by frequency distributions to validate the plausible ranges of GIs [11,40,53]. Similarly, graphical evaluation of the normalization, weighting, and aggregation procedures was used here to evaluate GI robustness.

This study considers a normal distribution for the different GIs developed for each TGPU. Thus, Chauvenet's criterion, detailed in [54], was used to identify spurious GIs considering a 95% probability. The deviation of each GI from the mean value of the GIs for the target TGPU is calculated and then divided by the standard deviation of the GIs of the TGPU. Then, the ratio is compared to a reference value, which depends on the number of GIs. The GI is spurious when the calculated ratio is higher than the reference value. The frequency distribution is plotted from the remaining GIs to confirm the robustness of the GI by graphical analysis.

*3.4. Visualization of the Developed GIs and Return to the Data*

From the plausible GIs, homogeneous groupings of TGPUs were made according to their characteristics. From the groups, the process returns to the source data (indicators) to evaluate the performance of the groups considering the performance objectives of cost, quality, flexibility, reliability, and speed. The grouping of TGPUs was made using a cluster analysis (CLA).

CLA is a multivariate data analysis approach with the purpose of group identification. It is a set of algorithms for classifying objects, such as countries, species, and individuals [2]. Classification reduces the dimensionality of a data set, exploring the similarities and differences so that similar elements are placed in the same group or cluster. A distance function is used to define the similarity or difference between the elements, which is defined considering the context of the problem being studied. CLA techniques can be hierarchical if the (increasing) number of classes is defined during the classification process itself, or non-hierarchical when the number of clusters is initially defined.

Among the existing procedures used to determine the distance between sets of observations, Ward's method [55] was applied in this study. In this technique, at each stage of the process, the two groups whose merging generates the minimal increase in variance are clustered. Another issue lies in the identification of the optimal number of clusters, which is largely subjective [2]. As an example of the application of this method, hierarchical cluster analysis was used to identify the similarities and differences in the sustainability performance of the 27 European Union countries [15]. The authors used a dendrogram to visualize the clustering process and, after evaluating where the distance values changed considerably, they suggested a certain number of clusters.

## 4. Table Grape Production Units (TGPUs), GIs, and Sustainability

*4.1. Table Grape Production Units (TGPUs)*

The production and global consumption of table grapes has increased in recent years due to the growing availability of the product in the market, increased consumer income, and changes in eating habits toward healthier products [56]. In this competitive market, table grape producers must demonstrate high operational efficiency [57]. A critical operation in this supply chain is the packaging of the table grapes because of its seasonality and intensive use of labor [58]. Packaging is carried out in the TGPUs, which are environmentally controlled, spacious, clean, and protected from sun, insects and animals. To better understand the objective of this work, Figure 3 depicts the post-harvest processing of table grapes, including the steps performed in a TGPU. The method proposed in this study for the development of robust GIs for performance evaluation was applied to the packaging process (highlighted in Figure 3).

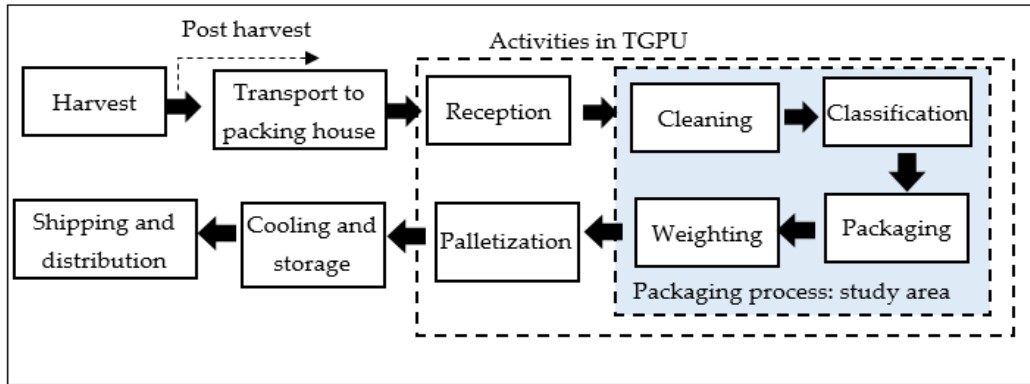

**Figure 3.** Post-harvest processing of table grapes-packaging process (study area) highlighted.

*4.2. Application of GIs in Sustainability Research*

Most of the work involving GIs have addressed issues related to sustainability, for example: Sustainable performance of wastewater treatment systems [39]; watersheds in

inter-basin water transfer projects [28]; and changes to the islands of the North Aegean region over time [59]. These studies developed the GIs by integrating indicators related to social, economic, and environmental factors. Similarly, these three themes are present in the sustainability assessment of TGPUs. In addition to the economic and social impacts related to workers, TGPUs affect the environment via waste disposal and the use of natural resources, such as water for the cleaning process and electricity for the air conditioning of the facilities. The GIs used here are expected to help in minimizing the environmental impact of TGPUs by indicating the factors that could increase the process efficiency (and minimize the use of natural resources) and increase the quality of the product (resulting in fewer off-spec products and less waste). The GIs used in the performance evaluation of TGPUs can be used to guide companies toward a sustainable business model (SBM).

According to [60], a SBM must develop an internal culture and structural capacity for the company to achieve sustainability, as well as collaborate with stakeholders so that the entire supply chain achieves sustainability. Among agricultural studies, the work of [61] is particularly relevant as the authors investigated how stakeholders in dairy cooperatives can contribute to innovation processes toward sustainability. Another study [62] investigated how the promotion of a sustainable culture in a food and beverage packaging company affects the actors in the supply chain. Finally, the influence of the sustainable and proactive behavior of a winery in encouraging the stakeholders to innovate and create business value was studied [63].

Here, the development of robust GIs for performance evaluation also contributes to a sustainable business model for TGPUs. The following hypotheses are formulated for the developed GIs:

- Null hypothesis: the proposed method allows the selection of a plausible range of robust GIs for the performance evaluation of TGPUs;
- Alternative hypothesis: the proposed method does not allow the selection of a plausible range of robust GIs for the performance evaluation of TGPUs.

## 5. Vale do Submédio São Francisco (VSSF; San Francisco Lower Middle Valley in Brazil) Case Study

First, the grape production of the VSSF region is contextualized to support the case study. Subsequently, the development of the case study and its respective results are presented.

### 5.1. VSSF Grape Production

Due to investments in infrastructure and the introduction of modern irrigation technologies, new crops and cultivation techniques, the VSSF is considered one of the most important agricultural regions in Brazil. One of the highlights of this region is the production of table grapes for export. Table grapes are the third most exported fresh fruit from Brazil, and almost all of Brazil's table grape exports are produced in the VSSF. Two main factors contribute to their competitiveness. First, the growing demand from the international market for fresh fruit, which expanded production in the large VSSF region, and second the opportunity to offer grapes during periods of low competition in Europe. However, concerns regarding the cost reduction of table grape production in the VSSF are important because of two immediate threats: (i) The entry of Peru in this market with low labor costs, favorable climate, and international investors promoting technological advances; (ii) the few international agreements to facilitate the entry of Brazilian grapes into other countries.

### 5.2. Development of a Robust GI

The steps carried out during the development of robust GIs for evaluating the performance of the TGPU production process in the VSSF are discussed here. Initially, the indicators, in their different units of measurement related to each performance objective

used in the GI models, were taken from a previous study [18], as summarized in Table 5. Labels A to M designate the thirteen TGPUs analyzed.

**Table 5.** Performance indicators for the TGPUs in the VSSF.

| TGPU | C_I | Q_I | F_I | R_I | S_I |
|------|-----|-----|-----|-----|-----|
| A | 0.745 | 0.932 | 1.153 | 76.6 | 116 |
| B | 0.769 | 0.917 | 0.697 | 79.4 | 85 |
| C | 1.000 | 0.831 | 1.642 | 91.1 | 155 |
| D | 1.000 | 0.820 | 2.230 | 90.2 | 191 |
| E | 0.606 | 0.869 | 0.318 | 99.4 | 87 |
| F | 0.742 | 0.882 | 1.815 | 86.2 | 145 |
| G | 0.629 | 1.000 | 1.857 | 89.0 | 91 |
| H | 0.657 | 0.938 | 0.955 | 90.4 | 73 |
| I | 1.000 | 0.373 | 1.992 | 89.7 | 226 |
| J | 1.000 | 0.000 | 1.931 | 89.1 | 255 |
| K | 0.962 | 0.883 | 1.920 | 89.5 | 110 |
| L | 1.000 | 0.892 | 2.333 | 88.1 | 81 |
| M | 0.493 | 0.969 | 1.371 | 87.0 | 52 |

C_I = cost indicator (dimensionless); Q_I = quality indicator (dimensionless); F_I = flexibility indicator (cycles/day); R_I = reliability indicator (%); S_I = speed indicator [kg/(benches·day)].

The different normalization/aggregation procedures described by the equations shown in Tables 2 and 4 were combined with equal weighting, resulting in the first nine alternative GIs from the possible fifteen combinations. Geometric aggregations were only possible with the Borda count normalization, because the other (four) methods contain null or negative values. Moreover, only Borda count, min-max, and relative to the maximum normalizations were considered in concave aggregations because their indicators are in the range between 0 and 1, which is not the case for the other (two) methods.

Then, the same possibilities of normalization/aggregation procedures were combined with weightings defined by five specialists (TGPU engineers and managers) using the equations shown in Table 3, resulting in nine more GIs. Table 6 shows the evaluations agreed between the participant experts (through meetings and discussions among them) and the weights obtained by the AHP method, in which the highest weight was for Q_I (0.33), the quality indicator, and immediately after, for C_I (0.30), the cost indicator.

**Table 6.** Pairwise comparison matrix based on AHP and resulting weights and consistency.

|  | C_I | Q_I | F_I | R_I | S_I | Eigenvector | Weights |
|------|-----|-----|-----|-----|-----|-------------|---------|
| C_I | 1 | 1 | 3 | 2 | 3 | 1.78 | 0.30 |
| Q_I | 1 | 1 | 3 | 3 | 3 | 1.93 | 0.33 |
| F_I | 0.33 | 0.33 | 1 | 0.33 | 1 | 0.52 | 0.09 |
| R_I | 0.50 | 0.33 | 3 | 1 | 3 | 1.08 | 0.19 |
| S_I | 0.33 | 0.33 | 1 | 0.33 | 1 | 0.52 | 0.09 |
| Sum $\geq$ | 3.17 | 3.00 | 11.00 | 6.67 | 11.00 | 5.83 | 1 |
| $n = 5$ | | $\lambda = 5.19$ | | $RI = 1.11$ | | $CR = 0.04 < 0.1$ (acceptable) | |

Table 7 shows the results of the 18 GIs obtained and further normalized between 0 and 1 in order to facilitate comparisons between different GIs and different TGPUs, as the developed GIs have different value scales.

In order to further facilitate visualization, the TGPUs are summarized in Table 8 according to their rank obtained. Most TGPUs had homogeneous classifications, without major variations.

**Table 7.** GIs of the TGPUs [0, 1].

| GI⁠/TGPU | 1 | 2 | 3 | 4 | 5 | 6 | 7 | 8 | 9 | 10 | 11 | 12 | 13 | 14 | 15 | 16 | 17 | 18 |
|---|---|---|---|---|---|---|---|---|---|---|---|---|---|---|---|---|---|---|
| A | 0.23 | 0.09 | 0.11 | 0.30 | 0.31 | 0.27 | 0.26 | 0.11 | 0.34 | 0.30 | 0.06 | 0.18 | 0.49 | 0.34 | 0.34 | 0.33 | 0.21 | 0.56 |
| B | 0.04 | 0.00 | 0.00 | 0.11 | 0.10 | 0.17 | 0.06 | 0.00 | 0.13 | 0.20 | 0.07 | 0.18 | 0.41 | 0.21 | 0.36 | 0.23 | 0.21 | 0.47 |
| C | 0.85 | 0.81 | 0.80 | 0.75 | 0.73 | 0.91 | 0.86 | 0.81 | 0.77 | 1.00 | 0.91 | 0.92 | 0.89 | 0.83 | 0.96 | 0.99 | 0.93 | 0.90 |
| D | 1.00 | 1.00 | 1.00 | 1.00 | 1.00 | 1.00 | 1.00 | 1.00 | 1.00 | 0.98 | 1.00 | 1.00 | 1.00 | 1.00 | 0.86 | 0.96 | 1.00 | 1.00 |
| E | 0.12 | 0.28 | 0.16 | 0.00 | 0.00 | 0.11 | 0.12 | 0.15 | 0.00 | 0.00 | 0.36 | 0.28 | 0.26 | 0.03 | 0.11 | 0.00 | 0.29 | 0.32 |
| F | 0.27 | 0.53 | 0.50 | 0.60 | 0.59 | 0.46 | 0.32 | 0.51 | 0.63 | 0.06 | 0.44 | 0.46 | 0.63 | 0.53 | 0.39 | 0.12 | 0.51 | 0.69 |
| G | 0.50 | 0.46 | 0.39 | 0.49 | 0.40 | 0.60 | 0.53 | 0.40 | 0.51 | 0.63 | 0.42 | 0.39 | 0.60 | 0.41 | 0.65 | 0.64 | 0.41 | 0.65 |
| H | 0.31 | 0.24 | 0.17 | 0.17 | 0.11 | 0.34 | 0.32 | 0.17 | 0.19 | 0.57 | 0.29 | 0.28 | 0.41 | 0.18 | 0.59 | 0.58 | 0.32 | 0.47 |
| I | 0.92 | 0.79 | 0.80 | 0.72 | 0.86 | 0.85 | 0.92 | 0.81 | 0.72 | 0.84 | 0.52 | 0.61 | 0.44 | 0.50 | 0.64 | 0.81 | 0.64 | 0.49 |
| J | 0.81 | 0.64 | 0.67 | 0.52 | 0.78 | 0.61 | 0.80 | 0.66 | 0.48 | 0.64 | 0.13 | 0.30 | 0.00 | 0.11 | 0.30 | 0.61 | 0.28 | 0.00 |
| K | 0.62 | 0.73 | 0.72 | 0.71 | 0.62 | 0.83 | 0.67 | 0.72 | 0.72 | 0.65 | 0.84 | 0.85 | 0.88 | 0.76 | 0.90 | 0.72 | 0.86 | 0.89 |
| L | 0.73 | 0.76 | 0.76 | 0.77 | 0.62 | 0.74 | 0.74 | 0.75 | 0.76 | 1.00 | 0.88 | 0.90 | 0.94 | 0.80 | 1.00 | 1.00 | 0.89 | 0.94 |
| M | 0.00 | 0.09 | 0.01 | 0.12 | 0.04 | 0.00 | 0.00 | 0.00 | 0.14 | 0.08 | 0.00 | 0.00 | 0.27 | 0.00 | 0.00 | 0.06 | 0.00 | 0.34 |

GIs [1–9 with equal weighting; 10–18 with weightings defined by specialists ($C\_I = 0.30$; $Q\_I = 0.33$; $F\_I = 0.09$; $R\_I = 0.19$; $S\_I = 0.09$)]:
1, 10 = Borda count/linear; 2, 11 = Z-scores/linear; 3, 12 = min-max/linear; 4, 13 = maximum/linear; 5, 14 = median/linear; 6, 15 = Borda count/geometric; 7, 16 = Borda count/concave; 8, 17 = min-max/concave; 9, 18 = maximum/concave.

**Table 8.** Ranking of the GIs in the TGPUs.

| GI⁠/TGPU | 1 | 2 | 3 | 4 | 5 | 6 | 7 | 8 | 9 | 10 | 11 | 12 | 13 | 14 | 15 | 16 | 17 | 18 |
|---|---|---|---|---|---|---|---|---|---|---|---|---|---|---|---|---|---|---|
| A | 10° | 11° | 11° | 9° | 9° | 10° | 10° | 11° | 9° | 9° | 12° | 11° | 7° | 8° | 10° | 9° | 12° | 7° |
| B | 12° | 13° | 13° | 12° | 11° | 11° | 12° | 13° | 12° | 10° | 11° | 12° | 10° | 9° | 9° | 10° | 11° | 9° |
| C | 3° | 2° | 3° | 3° | 4° | 2° | 3° | 2° | 2° | 1° | 2° | 2° | 3° | 2° | 2° | 2° | 2° | 3° |
| D | 1° | 1° | 1° | 1° | 1° | 1° | 1° | 1° | 1° | 3° | 1° | 1° | 1° | 1° | 4° | 3° | 1° | 1° |
| E | 11° | 9° | 10° | 13° | 13° | 12° | 11° | 10° | 13° | 13° | 8° | 10° | 12° | 12° | 12° | 13° | 9° | 12° |
| F | 9° | 7° | 7° | 6° | 7° | 8° | 9° | 7° | 6° | 12° | 6° | 6° | 5° | 5° | 8° | 11° | 6° | 5° |
| G | 7° | 8° | 8° | 8° | 8° | 7° | 7° | 8° | 7° | 7° | 7° | 7° | 6° | 7° | 5° | 6° | 7° | 6° |
| H | 8° | 10° | 9° | 10° | 10° | 9° | 8° | 9° | 10° | 8° | 9° | 9° | 9° | 10° | 7° | 8° | 8° | 10° |
| I | 2° | 3° | 2° | 4° | 2° | 3° | 2° | 3° | 4° | 4° | 5° | 5° | 8° | 6° | 6° | 4° | 5° | 8° |
| J | 4° | 6° | 6° | 7° | 3° | 6° | 4° | 6° | 8° | 6° | 10° | 8° | 13° | 11° | 11° | 7° | 10° | 13° |
| K | 6° | 5° | 5° | 5° | 6° | 4° | 6° | 5° | 5° | 5° | 4° | 4° | 4° | 4° | 3° | 5° | 4° | 4° |
| L | 5° | 4° | 4° | 2° | 5° | 5° | 5° | 4° | 3° | 2° | 3° | 3° | 2° | 3° | 1° | 1° | 3° | 2° |
| M | 13° | 12° | 12° | 11° | 12° | 13° | 13° | 12° | 11° | 11° | 13° | 13° | 11° | 13° | 13° | 12° | 13° | 11° |

Despite the homogeneous classifications observed in Table 8, and as proposed before, the Chauvenet test was applied to identify outlier GIs for each TGPU, and the results are summarized in Table 9. Four outliers were observed, three in the first round of verification (O1) and one in the second round (O2) (no outliers were observed in the third round of verification).

The initial test considered a 95% probability for the sample distribution of the 18 GIs for the TGPUs. In the first test, GI 5, GI 10, and GI 15 were identified as outliers. The second test was performed with the 15 remaining GIs and GI 16 was identified as an outlier. Then, the third test was performed with the 14 remaining GIs and no other outlier was identified. Thus, from the 14 remaining alternatives a graph is presented to visualize the distribution of the 13 TGPUs according to their rank in Figure 4. In this figure the TGPUs are presented following the rank obtained and, in addition, the stronger the areas shaded in red, the more certain its position in the classification. A rather narrow range of ranks of the TGPUs was identified by the graphical observation of the frequency distribution. For example, TGPU D was clearly ranked first. Thus, after Chauvenet's test was applied to indicate outlier results (Table 9) and the graphical evaluation of the rank distribution of the remaining GIs (Figure 4), the null hypothesis in this study is confirmed because the proposed method allowed the selection of a plausible range of robust GIs for the 13 TGPUs of the VSSF.

**Table 9.** Identification of outlier GIs.

| GI / TGPU | 1 | 2 | 3 | 4 | 5 | 6 | 7 | 8 | 9 | 10 | 11 | 12 | 13 | 14 | 15 | 16 | 17 | 18 |
|---|---|---|---|---|---|---|---|---|---|---|---|---|---|---|---|---|---|---|
| A | ✓ | ✓ | ✓ | ✓ | ✓ | ✓ | ✓ | ✓ | ✓ | ✓ | ✓ | ✓ | ✓ | ✓ | ✓ | ✓ | ✓ | ✓ |
| B | ✓ | ✓ | ✓ | ✓ | ✓ | ✓ | ✓ | ✓ | ✓ | ✓ | ✓ | ✓ | ✓ | ✓ | ✓ | ✓ | ✓ | ✓ |
| C | ✓ | ✓ | ✓ | ✓ | O1 | ✓ | ✓ | ✓ | ✓ | ✓ | ✓ | ✓ | ✓ | ✓ | ✓ | ✓ | ✓ | ✓ |
| D | ✓ | ✓ | ✓ | ✓ | ✓ | ✓ | ✓ | ✓ | ✓ | ✓ | ✓ | ✓ | ✓ | ✓ | O1 | O2 | ✓ | ✓ |
| E | ✓ | ✓ | ✓ | ✓ | ✓ | ✓ | ✓ | ✓ | ✓ | ✓ | ✓ | ✓ | ✓ | ✓ | ✓ | ✓ | ✓ | ✓ |
| F | ✓ | ✓ | ✓ | ✓ | ✓ | ✓ | ✓ | ✓ | ✓ | O1 | ✓ | ✓ | ✓ | ✓ | ✓ | O2 | ✓ | ✓ |
| G | ✓ | ✓ | ✓ | ✓ | ✓ | ✓ | ✓ | ✓ | ✓ | ✓ | ✓ | ✓ | ✓ | ✓ | O1 | ✓ | ✓ | ✓ |
| H | ✓ | ✓ | ✓ | ✓ | ✓ | ✓ | ✓ | ✓ | ✓ | ✓ | ✓ | ✓ | ✓ | ✓ | ✓ | ✓ | ✓ | ✓ |
| I | ✓ | ✓ | ✓ | ✓ | ✓ | ✓ | ✓ | ✓ | ✓ | ✓ | ✓ | ✓ | ✓ | ✓ | ✓ | ✓ | ✓ | ✓ |
| J | ✓ | ✓ | ✓ | ✓ | ✓ | ✓ | ✓ | ✓ | ✓ | ✓ | ✓ | ✓ | ✓ | ✓ | ✓ | ✓ | ✓ | ✓ |
| K | ✓ | ✓ | ✓ | ✓ | ✓ | ✓ | ✓ | ✓ | ✓ | ✓ | ✓ | ✓ | ✓ | ✓ | ✓ | ✓ | ✓ | ✓ |
| L | ✓ | ✓ | ✓ | ✓ | ✓ | ✓ | ✓ | ✓ | ✓ | ✓ | ✓ | ✓ | ✓ | ✓ | ✓ | ✓ | ✓ | ✓ |
| M | ✓ | ✓ | ✓ | ✓ | ✓ | ✓ | ✓ | ✓ | ✓ | ✓ | ✓ | ✓ | ✓ | ✓ | ✓ | ✓ | ✓ | ✓ |

| TGPU | 1st | 2nd | 3rd | 4th | 5th | 6th | 7th | 8th | 9th | 10th | 11th | 12th | 13th |
|---|---|---|---|---|---|---|---|---|---|---|---|---|---|
| D | 14 | 0 | 0 | 0 | 0 | 0 | 0 | 0 | 0 | 0 | 0 | 0 | 0 |
| C | 0 | 8 | 6 | 0 | 0 | 0 | 0 | 0 | 0 | 0 | 0 | 0 | 0 |
| L | 0 | 3 | 5 | 3 | 3 | 0 | 0 | 0 | 0 | 0 | 0 | 0 | 0 |
| I | 0 | 3 | 3 | 2 | 3 | 1 | 0 | 2 | 0 | 0 | 0 | 0 | 0 |
| K | 0 | 0 | 0 | 7 | 5 | 2 | 0 | 0 | 0 | 0 | 0 | 0 | 0 |
| F | 0 | 0 | 0 | 0 | 3 | 5 | 3 | 1 | 2 | 0 | 0 | 0 | 0 |
| G | 0 | 0 | 0 | 0 | 0 | 2 | 8 | 4 | 0 | 0 | 0 | 0 | 0 |
| J | 0 | 0 | 0 | 2 | 0 | 4 | 1 | 2 | 0 | 2 | 1 | 0 | 2 |
| H | 0 | 0 | 0 | 0 | 0 | 0 | 0 | 3 | 6 | 5 | 0 | 0 | 0 |
| A | 0 | 0 | 0 | 0 | 0 | 0 | 2 | 1 | 2 | 3 | 4 | 2 | 0 |
| E | 0 | 0 | 0 | 0 | 0 | 0 | 0 | 1 | 2 | 3 | 2 | 4 | 2 |
| B | 0 | 0 | 0 | 0 | 0 | 0 | 0 | 0 | 2 | 1 | 3 | 5 | 3 |
| M | 0 | 0 | 0 | 0 | 0 | 0 | 0 | 0 | 0 | 0 | 4 | 3 | 7 |

**Figure 4.** Distribution of GI rank frequencies among the 13 TGPUs.

*5.3. Visualization of the GIs and Return to the Data*

Based on these plausible GIs, groups were classified to facilitate decision-making to improve the performance of the TGPUs. In the first experiment the TGPUs were grouped using CLA. A hierarchical analysis was made to group the TGPUs according to similarities in the GIs obtained, as well as this, the grouping was adopted in three performance classes (low, medium, and high) (Figure 5).

From the identified groups we returned to the individual indicators to analyze the performance of the groups for each performance objective (cost, quality, flexibility, reliability, and speed) shown in Table 5. The individual indicators adjusted to the same range of values [0, 1] were used to calculate the average of the different TGPUs groups and to produce the spider graph shown in Figure 6. It is observed that if we had to adopt only one GI this could be that one whose rank is the most correlated with the median rank among the 14 plausible GIs. In this case it would be GI 12 (normalization [0, 1], weightings defined by specialists and linear aggregation) with a correlation of 0.99. The groups showed different performances. Group 1 (C, D, I, K, L) contains the five highest-ranked GIs and had

a better global performance than the others. However, it obtained the lowest evaluation for the quality objective. Group 3 (A, H, E, B, M) had the five lowest-ranked GIs and had the lowest overall evaluation. Group 2 (F, G, J) showed intermediate rankings. Thus, the proposed method can be valuable for the TGPU managers to identify the best references by groups and to study the good practices in order to implement them in their TGPUs. This would make the planning of actions for the development of the TGPU better. In addition, the set of developed GIs or a specific GI (for example GI 12) together with their constituent indicators can be used for the continuous monitoring and improvement of the operation of the TGPUs. These indicators can also be used by public managers in drawing up government plans and policies, contributing to the sustainable development of the industry with respect to social, economic, and environmental issues, for example in local productive arrangements.

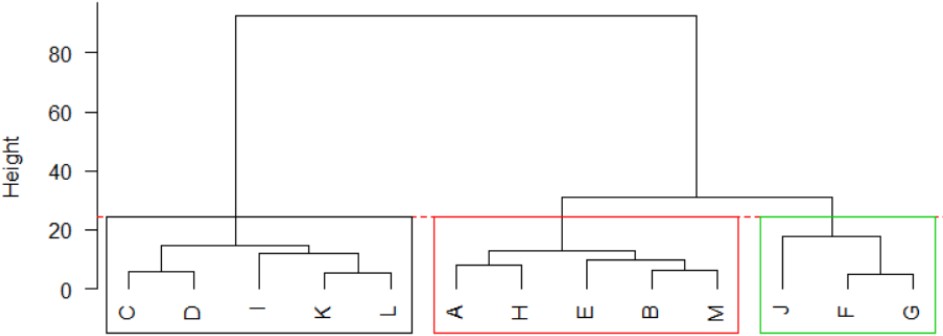

**Figure 5.** Dendrogram describing the classification of the TGPU groups by cluster analysis.

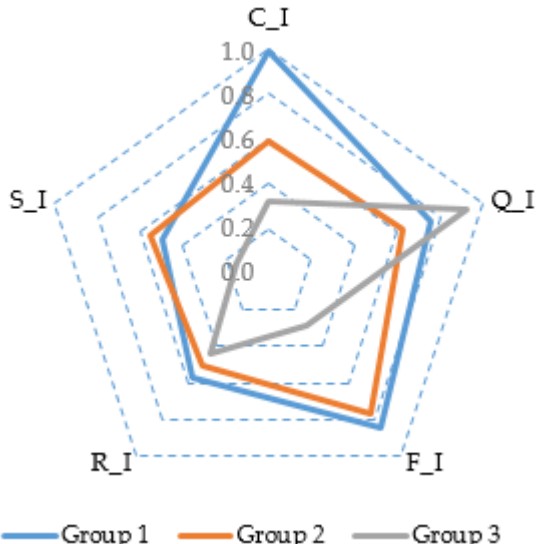

**Figure 6.** Groups performance per performance objectives.

## 6. Conclusions

The proposition of the method for the development of a robust GI for TGPUs, according to the scheme shown in Figure 1, included the following steps: collection of indicators by performance objective; establishment of GIs by means of different combinations of procedures (normalization, weighting and aggregation); identification of outlier GIs; definition of plausible GIs; evaluation of TGPUs by group and performance objective.

The proposed method was effective for developing plausible GIs for evaluating the performance of the case study TGPUs. The plausible GIs developed by the different normalization, weighting, and aggregation techniques were defined by identifying outlier GIs and analyzing the distribution of the ranks of each TGPU. CLA of the GIs was used to

classify the TGPUs into groups, which were then evaluated for each performance objective. Therefore the gap in the literature regarding the development of GIs for performance evaluation of TGPU production processes, essential in the table grape production chain, has been successfully addressed. Moreover, the identification of outlier GIs and the grouping of plausible GIs add a new scientific approach to this topic. These are the main contributions of this research.

The findings of this study are expected to contribute to the development of robust GIs for evaluating and comparing the performance of TGPUs and other similar scenarios. The comparison stimulates practices related to the concept of benchmarking, which are essential for the development of the VSSF and other similar regions. Another practical implication of this proposal is the use of GIs in shaping decision-making and (public and private) policies toward sustainability of TGPUs and the entire supply chain. All of these are expected to promote overall sustainability in the agricultural industry.

Considering the potential for the practical application of the method, which compares the performance of similar production units, it is believed that it can be extended to other agricultural products, such as cheese, milk, coffee, and wine, among others. The producers of such products are often organized into cooperatives, which often prioritize the sustainable development of all parties through cooperation.

A possible limitation in the application of the method to other processes concerns the collection of performance indicators, because of the need to use indicators that adopt the same criteria in their development. However, this limitation can be overcome by involving unit managers in the planning of data collection. Although this study is oriented to the evaluation of an agricultural product, the proposed method can be adjusted and applied to other areas. In future research, the proposed method could be promising for the formation of GIs to evaluate the sustainability of the 27 federative units (FUs) of Brazil. The Brazilian Institute of Geography and Statistics (IBGE) Automatic Recovery System (SIDRA) makes numerous sustainability indicators available for the Brazilian FUs with environmental, social, economic, and institutional themes. Classifying the FUs using CLA would make it possible to observe the performance of the groups in each theme and thus promoting policies in favor of the sustainable progress of the FUs and consequently of the country.

**Author Contributions:** Conceptualization, E.K., A.F. and C.C.; methodology, E.K., A.F. and C.C.; validation, E.K. and A.F.; formal analysis, M.E.; investigation, E.K.; resources, E.K.; data curation, E.K. and C.C.; writing—original draft preparation, E.K.; writing—review and editing, M.E.; supervision, M.E. All authors have read and agreed to the published version of the manuscript.

**Funding:** This research received no external funding.

**Institutional Review Board Statement:** Not applicable.

**Informed Consent Statement:** Not applicable.

**Data Availability Statement:** The data presented in this study are available on request from the corresponding author. The data are not publicly available due to restrictions privacy.

**Conflicts of Interest:** The authors declare no conflict of interest.

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
