# Peer review of "Development of Performance Evaluation Indicators for Table Grape Packaging Units. 2. Global Indexes"

_sustainability, doi:10.3390/su13116367_

Round 1

Reviewer 1 Report

P.2 row 4: it seems a verb is missing after "GI"

The rest of the work is well structured and very well written. I would have suggested accepting it as it is, if there were not this small, meaningless missing verb (a typo, surely)

Author Response

Reviewer 1

“P.2 row 4: it seems a verb is missing after "GI"

The rest of the work is well structured and very well written. I would have suggested accepting it as it is, if there were not this small, meaningless missing verb (a typo, surely)”

            We thank the reviewer for his/her encouragement words and positive assessment of our work. We added the word "should" in P.2 row 4. So the new sentence in lines 47-49 is:

They have proposed that the robustness of the GI should be assessed considering how the selection of the input procedures propagates through the structure of the GI and affects the results.

Reviewer 2 Report

  1. Need to highlight the contribution of this research.
  2. How to develop the method for a robust GI for TGPUs is illustrated?

Author Response

Reviewer 2

“2. Global indexes background and methods

We suggest 2 separate paragraphs for Global indexes background, and methods. The authors approach make difficult text analyzing, which is not presented in a clear manner. Methods, theoretical considerations upon global indexes and table grape production units and study hypothesis are mixed in text, and from here the difficulty to follow the ”red file” of the exposed issues.”

            The original section “2. Global indexes background and methods” was already divided into several paragraphs and several subsections. Nevertheless, and considering that the reviewer wants to refer to dividing it into other main sections, we have separated this section into three other main sections, and we thank the reviewer for helping us to make our paper even more clear. Thus the new structure of the manuscript follows as indicated in lines 71-75:

The background of GIs and the methods used here are presented in Sections 2 and 3, respectively. A description of the TGPUs, the application of GIs for sustainability topics and the study hypotheses are shown in Section 4. The case study of VSSF grape production and the calculations of the GIs are discussed in Section 5. Finally, conclusions and suggestions for future work are provided in Section 6.

“Please give a brief, but clear, description of the proposed method (that is the result of your study), before formulating conclusions. We suggest to use a schematic approach.”

            We added the following paragraph in lines 461-465:

The proposition of the method for the development of a robust GI for TGPUs, according to the scheme shown in Figure 1, included the following steps: collection of indicators by performance objective; establishment of GIs by means of different combinations of procedures (normalization, weighting and aggregation); identification of outlier GIs; definition of plausible GIs; evaluation of TGPUs by group and performance objective.

Reviewer 3 Report

  1. Global indexes background and methods

We suggest 2 separate paragraphs for Global indexes background, and methods. The authors approach make difficult text analyzing, which is not presented in a clear manner. Methods, theoretical considerations upon global indexes and table grape production units and study hypothesis are mixed in text, and from here the difficulty to follow the ”red file” of the exposed issues.

Please give a brief, but clear, description of the proposed method (that is the result of your study), before formulating conclusions. We suggest to use a schematic approach.

Author Response

(The authors gave the same response as above.)

Round 2

Reviewer 2 Report

  1. How to  develop the  robust GI for TGPUs?
  2. Need to enhance the contribution of this research.

Author Response

Reviewer 2

“1. How to develop the robust GI for TGPUs?”

            The method for developing the robust GI for TGPUs is illustrated in Figure 1 and its description is presented in lines 461-465:

The proposition of the method for the development of a robust GI for TGPUs, according to the scheme shown in Figure 1, included the following steps: collection of indicators by performance objective; establishment of GIs by means of different combinations of procedures (normalization, weighting and aggregation); identification of outlier GIs; definition of plausible GIs; evaluation of TGPUs by group and performance objective.

“2. Need to enhance the contribution of this research.”

            The contribution of this research is presented in lines 471-475, and it is highlighted in this new version:

Therefore the gap in the literature regarding the development of GIs for performance evaluation of TGPU production processes, essential in the table grape production chain, has been successfully addressed. Moreover, the identification of outlier GIs and the grouping of plausible GIs adds a new scientific approach to this topic. These are the main contributions of this research.

Reviewer 3 Report

The paper may be published in the present form.

Author Response

Reviewer 3

“The paper may be published in the present form.”

            We thank the reviewer for his/her encouragement words and positive assessment of our work.
